# Determining the Factors That Influence Stunting during Pandemic in Rural Indonesia: A Mixed Method

**DOI:** 10.3390/children9081189

**Published:** 2022-08-08

**Authors:** Esti Yunitasari, Bih O. Lee, Ilya Krisnana, Rayi Lugina, Fitriana Kurniasari Solikhah, Ronal Surya Aditya

**Affiliations:** 1Faculty of Nursing, Universitas Airlangga, Surabaya 60115, Jawa Timur, Indonesia; 2Nursing Department, Kaohsiung Medical University, Kaohsiung City 807, Taiwan; 3Nursing Department, Politeknik Kesehatan Kemenkes Malang, Malang 65119, Jawa Timur, Indonesia; 4Department of Public Health, Faculty of Sports Science, Universitas Negeri Malang, Malang 65145, Jawa Timur, Indonesia

**Keywords:** COVID-19, stunting, sanitation, child

## Abstract

Objective: Pandemic causes an increase in the poverty rate. The consequences will be many, including the birth of stunting babies. The COVID-19 pandemic has had an impact on stunting. Analyzing the factors that cause stunting during a pandemic will provide suggestions for effective stunting prevention strategies at the national, regional, community, and household levels. This study aims to determine the factors that influence stunting during the pandemic. Method: We use mixed methods. The respondents of this study were 152 mothers of the Maternal and Child Nutrition Security project, and the sampling technique is Cluster Sampling. Quantitatively using a baseline survey whose analysis uses multiple logistic regression to determine the unadjusted and adjusted odds ratio. The qualitative data used focus group discussions which were analyzed using Nvivo 12 with a questionnaire, and anthropometric measurements of children from surveyed households. Results: This study summarizes the multivariate analysis of stunting determinants in the pandemic era, revealing statistically significant interactions between household sanitation facilities and household water treatment. Significant risk factors for severe stunting during the pandemic were male gender, older child age, coming from a low socioeconomic quintile, not participating in prenatal care at a health facility, and mother’s involvement in choices about what to prepare for Community House. The FGDs identified misinformation about childcare and consumption of sweetened condensed milk as significant contributors to child malnutrition. Conclusions: Lack of sanitation facilities and untreated water are contributing factors. Water, sanitation, and hygiene initiatives must be included into Indonesian policies and programs to combat child stunting during a pandemic. The need for further research related to government assistance for improving toddler nutrition, as well as the relationship between WASH and linear development in early infancy should be explored.

## 1. Introduction

Communities around the world are taking steps to limit the spread of the acute respiratory syndrome coronavirus or Coronavirus Disease-2019 (COVID-19) and reduce the number of deaths caused by the virus [1]. Authorities should consider the direct health consequences of the pandemic and the indirect consequences and responses to the epidemic while assessing their choices [2]. Although the mortality rate for COVID-19 appears modest among children and women of reproductive age, the virus has been linked to several diseases such as dengue fever, tuberculosis, measles [3].

During the COVID-19 pandemic, we wanted to see what the indirect effects of stunting might look like. In addition, multi-sectoral operations have not been carried out optimally in Indonesia due to the lack of information about the causes of stunting in Indonesia during the pandemic which can be used as guidelines for designing multi-sectoral programs. Specifically on stunting, the World Health Organization (WHO) has published two policy briefs that guide how to accelerate progress toward global targets and combat stunting while ensuring equity [2], as action points to realize the stunting reduction agenda.

Since the pandemic period in 2020, the poverty rate has increased, of course there will be many further impacts, including the birth of stunting babies. Experts also say that the stunting rate from 27 percent will increase to 32 percent [4]. This shows that the COVID-19 pandemic does have an effect on stunting. The need for effective strategies for stunting prevention at the national, regional, community, and household levels is apparent, and it is essential to engage individuals from a variety of different sectors and disciplines in the process of developing them [5]. Because of the complex character of malnutrition and the necessity of multi-sectoral treatments, international policy recommendations consider this [4].

The high incidence of stunting in Indonesia, the numerous factors that influence the incidence of stunting, and the low level of health behavior toward the incidence of stunting have piqued the interest of researchers who wish to research the incidence of stunting during the COVID-19 pandemic [6]. The number of stunting cases in Indonesia in 2019 reached 27.67 percent. Stunting determined by the World Health Organization (WHO), Indonesia’s status is still 4th in the world and 2nd in Southeast Asia regarding cases of stunting under five [7], among other reasons. This study used the mix method, which is a combination of focus group data and data from a cross-sectional survey to examine maternally, child, and household factors associated with stunting and severe stunting in children, including household water, sanitation, and hygiene (WASH) indicators, facilities, and practices, to further investigate the relationship between the variables.

There is observed a high incidence of stunting in Indonesia and the multiple factors that influence it. The goal of this research is to figure out what factors influence stunting during a pandemic. Our findings suggest theater can increase awareness of implicit bias and encourage behavior change among health professionals and members of the public, but more is needed to help health care providers prevent stunting. Theater can increase awareness and empathy around stunting in children during a pandemic. The use of this theater can provide a platform for having conversations about critical prevention strategies in times of future pandemics

Stunting during a pandemic may be predicted using statistical modeling, which can help policymakers make informed choices. It has been possible to assess the direct effects of COVID-19 [8], including those on pregnant women and babies, via the use of models. Although it is still early in the epidemic, the research will offer a set of realistic and quantifiable estimates that will serve as a reference point for decision-makers who are presently contemplating response measures.

## 2. Material and Method

We performed the research between 9 and 31 March 2021. As an indigenous subsistence rural population, agricultural fields confront the difficulties of steep and rocky terrain and a dry seasonal environment (906 mm average annual rainfall). The mixed-methods technique comprises three components: (1) Discussion groups with caregiver and mothers; (2) distributing questionnaires to all homes without sample; and (3) anthropometric measurements of children without sampling. We work with nurses and the government as an intercultural team to design research tools that reflect the local environment and promote participant participation in the project. Nurses and the government selected the study subject (e.g., stunting) with feedback from several communities. We included these questions in the questionnaire.

### 2.1. Subjects

An Indonesian Maternal and Child Nutrition Security project baseline survey completed between 9 and 31 March 2021 was utilized as the basis for the study. Three unique West Java typologies were represented by the 14 districts chosen for this project: Stunting is prevalent in Pasirjati, a coastal area in West Java Province, Indonesia, and 152 mothers with infants aged 0–23 months comprise the research sample for this study.

Inclusion criteria:Mothers who have a child aged 0–23 monthsChildren aged 0–23 months who have MCH/KMS and are registered at the Puskesmas Pasirjati, Bandung.

Exclusion Criteria:Children who are accompanied or are experiencing co-morbidities such as diarrhea.Children with disorders such as autism and mental retardation.Children who have specific food allergies.

### 2.2. Sampling

Pasirjati is divided into 14 sections. The Puskesmas (Community Health Services) were chosen at random from each sub-district in the second step, and then villages were chosen at random from the Puskesmas in the third stage. Using a sample frame of every ten families, the closest household from the local health service (Pustu) was chosen as the beginning point for each cluster. There are a total of samples that have been counted, cluster 1 (9 samples), cluster 2 (12 samples), cluster 3 (12 samples), cluster 4 (10 samples), cluster 5 (10 samples), cluster 6 (12 samples), cluster 7 (14 samples), cluster 8 (10 samples), cluster 9 (11 samples), cluster 10 (9 samples), cluster 11 (11 samples), cluster 12 (10 samples), cluster 13 (13 samples) and cluster 14 (11 samples). The use of Cluster Sampling is intended so that each cluster has a representative. Sample selection was assisted by using a Random Number Generator (RNG) where the data had previously been sorted according to the research criteria and grouped based on the cluster where the respondent lived. This RNG is used to facilitate researchers in selecting respondents randomly.

### 2.3. Data Collection

Data on 0–23-month-old children, their mothers, and their households were gathered via the use of a standardized questionnaire. In addition to the child’s age, gender, weight, and height/length, additional factors to consider include the mother’s age, education level, and role in household decision-making, as well as her usage of breast milk, supplemental feeding, and handwashing (antenatal care, assistance during delving, and postnatal care). A locally made length board is used to precisely measure the length of a kid (for those aged 0 to 23 months). SECA Elektronik scales were used to measure the child’s weight and length (for children aged 0–23 months) using an electronic SECA Elektronik scale with a 0.1 kg accuracy. The SECA scales were calibrated with a standard weight of 5 kg each morning before data collection and then used to gather data. Repeated anthropometric measurements of 10% of the sample yielded a coefficient of variation less than 5% in children and women, respectively. All enumerators received at least two days of training before data collection, and those who were responsible for taking anthropometric measures received an additional day of instruction. It is the job of supervisors to ensure that enumerators are doing their job correctly and to build strong connections with the community people they serve.

### 2.4. Performing a Stunt (High-for-Age)

Anthropometric measurements were used to assess the nutritional condition of children under the age of five. Children less than two years old had their length measured, while those more than two years old had their height measured. A wooden stadiometer and a Microtoice tape were used to measure length and height, respectively, to within 0.1 cm. The standard deviation (SD) (Z-score) from the median of the reference population is used to indicate the current state of the measurement of height according to age. A kid was termed stunted if their standard deviation in height (SD) was more than or equal to two standard deviations below the median of the reference population.

### 2.5. Socio-Economic Aspects

A structured household questionnaire was used to collect data on the following family level factors: region (urban and rural), district (eight in total), father’s education level (completed primary school [6 years of schooling], completed secondary school [12 years of schooling], and completed secondary school [12 years of schooling]); mother’s education level; parental education (both with higher education, father with higher education, and mother with higher education); and children’s education level (completed primary school [6 years of schooling], completed. In addition, the following child-level factors were pooled for analysis: the child’s age in months, gender, the number of prenatal visits, and the supply of information on nutritional status during pregnancy. After obtaining consent to participate in the study verbally and in writing, the questionnaire was carried out. Every day, the supervisor in the field checked the questionnaires to ensure that they were accurate, consistent, and comprehensive.

### 2.6. Quantitative Analytical Statistics

A computerized database was created and sanitized using the EPIINFO data entry tool [9]. Data on nutritional status were analyzed using a new World Health Organization growth reference. The wealth index score was calculated based on household ownership of the aforementioned consumer goods using a method similar to that described by Filmer and Pritchett and then divided into three categories. The poorest are those in the bottom 40% of the income distribution, followed by the middle 40% and the poorest 20%. Examining children ranging in age from 0 to 23 months with mild, moderate, or severe stunting. A dichotomous variable (category 0 if not stunted (−2SD) or severely stunted (−3SD) and category 1 if stunted (−2SD) or highly stunted (−3SD)) was used to define the extent of stunting and severe stunting (−3SD).

Stunted and severely stunted 0–23-month-old children were first analyzed using a univariate binary logistic regression analysis. After that, the factors linked to stunting and severe stunting were examined using a multivariate logistic regression model. A step-by-step technique to reverse elimination is adopted. It was decided to include the initial variable in the model if the univariate *p* value fell below or was equal to 0.25. In the final model, only the parameters that were substantially related with stunted and highly stunted children (*p* 0.05) remained. The unadjusted and adjusted odds ratios of the logistic model are displayed with 95% confidence intervals. For data analysis, the “SVY” command from Stata version 11 (Stata Corp., College Station, TX, USA) was used to adjust the cluster sampling design and sample weights suitably.

### 2.7. Qualitative Analytical Statistics

Group discussion with women from the community on 9 March and 31 March 2021. The goal of these discussion groups is to better understand community views to better guide the interpretation of quantitative data in the future. Aside from that, group talks are essential for including community members in a meaningful reflection to discover actionable aspects that will help drive data more thoroughly. The government and nurses recruited people of the community, starting with moms, who each had approximately 19 participants. Following the wishes of the participants, the debate was held in both Sundanese and Indonesian. Each session is moderated by a nurse who offers questions to stimulate conversation. The conversation starts with a single question that is asked. When government officials and nurses visited the women in their communities during the epidemic, they questioned them why they believed nutrition was an issue in their community. Before concluding, the group is asked to evaluate their significance on a scale of 1–5, with one being least essential and five being most significant, based on the issues brought up throughout their conversation. With the assistance of the research assistant, the author documented the conversation and the findings of the discussion of nutrition-related variables that occurred.

Two researchers used the content analysis technique to examine audio recordings of mothers discussing health-care services and the prevalence of stunting in their children. Three researchers classified the participants’ answers, and a third researcher used Nvivo 12 to generate an updated version of the data from the original data set. The transcripts from the FGDs were analyzed using Nvivo 12. Coding was reviewed by the study team, differences were discussed, relationships between themes were discussed, and new codes were created if deemed necessary. Triangulation of emerging findings was carried out on surveys, community FGDs, questionnaires, and anthropometric measurements to identify similarities and differences.

### 2.8. Data Input and Analysis Are Required

About 10% of the surveys were filled out incorrectly by the data entry operators, and fewer than 0.7% of the fields they entered had keyboard errors. Multiple checks were conducted to verify there were no duplications, outliers, or missing data in the datasets. Stata 11.0 was used for the statistical analysis. The xtgee function in Stata 11.0 was used to provide robust estimates after the data were modified for cluster sampling using the General Estimation Equation model. A lack of data on PMT and maternal access to prenatal and delivery care for children under the age of 23 months prompted the study to focus on children under that age. Stunting can be predicted by looking at this particular signal, which is why it was included in the research. A decision was made to exclude children with missing or erroneous data for any of the variables under examination from the analytical sample. When it comes to stunting and severe stunting, the percentage of children whose weight for height z-score was less than or equal to the mean minus standard deviation and standard deviation, respectively, of the World Health Organization (WHO) Child Growth Standards median height for age is the most important factor to consider. This research makes use of the WHO’s definition of IDM practice indicators in children under the age of two. An additional variable, referred to as “age-appropriate feeding,” takes into account both exclusive breastfeeding for babies younger than five months and the consumption of a minimally acceptable diet by children older than six months. A child can only be considered age-appropriate in the first two categories if she is either exclusively breastfed or has been fed by her mother exclusively since the age of five months. In the UNICEF and WHO Joint Monitoring Program, the operational definitions used for better drinking water supply treated water and improved sanitation are described here.

## 3. Results

For women and their families with babies ranging in age from 0 to 23 months, Table 1 provides data on the socioeconomic position. Moreover, 14 percent of the parents polled were under the age of 20 and had only finished the first grade of high school, according to the study. However, 61.6% of households lacked access to a modern toilet, and only 43.3% of those surveyed used proper disposal methods for children’s waste. Only 55.2% of households reported using soap to clean their hands. Most people (more than 90%) admitted to treating their tap water before drinking it despite just 31.2 percent of households having access to high-quality water sources. Pregnant women who got ANC from a doctor or a midwife and at a private or public health facility were 95.1 percent of the moms of the children they were caring for. 88% of women engaged in home food decisions, 89% in family meal preparation, 95% in child nutrition, and 86% in securing health care for their children. For newborns and young children, Table 2 provides information on their nutritional status and feeding techniques. The incidence of moderate and severe stunting combined was 29.5%, whereas the prevalence of severe stunting was 5.8%. Moreover half (55.2 percent) of infants under the age of six months were exclusively breastfed during the first hour of their lives. Mini-mom-acceptable diets that contained enough milk feeds, meals, and food categories only went to 35.7% of 6–23-month-olds, according to the CDC data. As described above, age-appropriate feeding was defined as nursing for children aged 0–5 months, and a minimum acceptable diet for children 6 to 23 months.

A bivariate and multivariate analysis of the link between overall stunting and child, mother, and household factors is summarized in Table 3 of this research. Stunting was found to be more common in children aged 12–23 months (38.9%) and 6–11 months (23.7%) than in children aged 0–5 months in the bivariate analysis (14.4 percent). Stunted children had a substantially higher risk of malnutrition than children who got the recommended daily amount of food for their age (30.2 percent vs. 23.1 percent). There is 95% certainty. Children in the lowest income quintile had two times the risk of being stunted as those in the highest income quintile, according to the study (AOR 2.31; 95 percent CI 1.43–3.68). Home sanitation facilities and water treatment (*p* for interaction 0.007) are also shown to have a significant impact on the likelihood of stunting in children. Stunting was previously not associated with poor latrine usage among children residing in treated water-using homes (adjusted odds ratio 1.26, 95% confidence interval: 0.99–1.63, *p* = 0.001). In the past, children who drank water that had been treated had a greater risk of being stunted.

We repeated the multivariate analysis on the dataset of children aged 0–23 months (*n* = 152) but omitted variables that were not present in the dataset of children aged 0–23 months (PMBA practices and maternal access to ANC during her last pregnancy). The AOR 1.29, 95 percent confidence interval 1.04–1.28 for stunting in households that treated their own water was found to be equally significant in the multiple logistic regression model (child gender, child age, wealth quintile), as was the interaction between household sanitation facilities and household water treatment (see Table 1), whereas poor sanitation in households that did not treat their own water was found to be equally significant (see Table 1) (AOR 1.28, 95 percent confidence interval 1.04–1.28 for stunting).

### 3.1. Focus Group Discussion

Table 4 shows the theme of the focus group discussion results from women who have stunting children and caregivers. The resulting themes are: Underestimating the importance of dietary supplements, having a fear of using dietary supplements, Frequent Shopping Doesn’t Matter, Children eat sweet food, Following the wrong Guide, Women’s involvement in policymaking, decision making, and control, Bathroom with adults and dirty, Absorption by the community, Provision of dietary supplements, When it comes to the supply of dietary supplements, there is occasionally a misunderstanding.

### 3.2. FGD with Women Who Have Children That Are Stunted

#### Underestimating the Importance of Dietary Supplements

Most caregivers do not consider stunting to be a health issue because they usually think that stunting is a genetic disease and that God has predetermined a person’s physical stature and height. Increased “physical strength” and “energy” are thought to be linked with supplementation in individuals of all ages, including children. Some caretakers believe that supplements have a beneficial effect on a kid’s health at birth, while others believe that supplements are just an extra bag of rations for the infant. Only if supplements are given free of charge and monitored regularly will the majority of carers be willing to take them, and only a small number of caregivers are prepared to be provided free modest quantities of these dietary supplements, with greater and lesser commitment obligations. Caregivers stated that they were interested in receiving free dietary supplements, but that it would be more beneficial to sell the supplements so that the money could be used to purchase other rations such as cooking oil and wheat, which could feed the entire family rather than a specific group of individuals. Participants indicated that they were not opposed to taking supplements if they were provided free of charge and that they required frequent monitoring, and that they were concerned that their kid would regress to a stunted state after the program ends.


*“Because the height of the parents influences the height of their offspring, it is normal for them to stun. Parents who are younger are more likely to have children.”*



*“The offspring will likewise be of a tiny stature. For pregnant and nursing moms, it is beneficial because it provides physical strength [energy].”*



*“Will this program continue once the healthy youngsters have graduated? If it does not persist, it is pointless.”*


### 3.3. Having a Fear of Using Dietary Supplements

Most mothers realize that a healthy family is the fulfillment of nutritional supplement needs. On the other hand, moms expressed concern about going to the health center for fear of catching the coronavirus. Because of housekeeping, they cannot take supplements on a regular basis and must instead rely on their children or spouse to get supplies. Husbands and children, on the other hand, often do not engage with nurses and ignore nurses’ advice regarding the usage and advantages of medications. Nursing staff dispensed goods in a rush, according to mothers, leaving them little time to learn why supplements should be taken. They also said that, despite the fact that nurses visited households, there was minimal monitoring of children’s growth and development, and there was little discussion regarding the use of nutritional supplements.

Because he said the supply was depleted and requested us to wait, I don’t get food supplements regularly.


*“Please give it a few days.”*



*“The usage and advantages of supplements, as well as other problems, should be discussed with him, according to the author.”*



*“We were also a little concerned when he came to visit since he worked at a health facility and might potentially carry the illness into our home.”*


### 3.4. Frequent Shopping Does Not Matter

Families with stunted children are generally concerned with the needs of adult families, including fathers, who are the main focus—while their father has a primary need, namely smoking. Smoking is a secondary need, but it consumes the most funds. Apart from that, the needs of other children, hence the baby gets the last priority for shopping for his needs. Therefore, there needs to be a solid effort to encourage people’s mindsets to reduce spending on non-essential things and focus on family nutrition to prevent stunting.


*“When shopping for necessities, his father always asks for cigarettes, so the needs for his children are somewhat marginalized.”*



*“My first child also often asks for snacks when shopping, so when we buy milk, we have less money.”*


### 3.5. Children Eat Sweet Food

Malnutrition, namely malnutrition and stunting, is due to errors in food intake, such as excessive sugar consumption (glucose). Children with high sugar intake can be at risk for experiencing growth disorders and poor nutrition. Sweetened condensed milk product advertisements that have been consumed by the public for years have resulted in people already assuming that sweetened condensed milk is milk that can be consumed by families. Sweetened condensed milk has even become one of the food items distributed by local governments, community institutions, and others to the community in the face of the COVID-19 Pandemic. The sugar content in sweetened condensed milk is almost 50 percent of the composition. There should be a limit to the amount of consumption allowed taking into account the age range


*“We can’t buy milk for babies, so the solution is Sweetened Condensed Milk. Moreover, we can get milk from government assistance because of the impact of COVID-19.”*



*“I often use sweetened condensed milk as an alternative for my child to drink milk.”*


### 3.6. Following the Wrong Guide

Most mothers rarely rarely read literature about the food to be given to their children. They have to follow the right guidelines. The instructions are issued by IDAI by WHO. They are more confident about the opinions of friends who are not necessarily true sources. In addition, misleading information from social media makes mothers confused. There is a need for health workers to provide a clear way to solve this problem because the first 1000 days require a balanced nutritional composition, with more protein in the diet.


*“It’s not because A’s mother said that her child would get fat faster by eating this. Or following Mother B, who said that home-cooked food is always healthier than boxed food.”*



*“I want to make a confession. I used to have fun choosing organic flour, which contains one kind of tuber to feed the baby. Imagine, babies are only given carbohydrates. When they actually need protein and other nutrients, why is that his son?”*



*“I am also a victim of naturalist mothers who prioritize vegetables and fruit for children rather than animal protein. The reason is that fruits and vegetables are easier for children to digest than meat, chicken, or fish. That is according to the understanding that I used to hear a lot from my friends.”*


### 3.7. Women’s Involvement in Policymaking, Decision Making, and Control

Strengthening family functions both through partnerships and paying attention to gender relations needs to be done. In addition, cooperation from various sectors is needed to resolve the issue of gender inequality, the issue of women and children, which are interrelated in overcoming the stunting problem. All parties encourage the development of gender-responsive family resilience as an effort to prevent stunting and build quality human resources—efforts through optimal care and nutrition in the golden period of children. Moreover, children are human resources (HR) who are also the future of a nation. The mother’s authority for decision-making is often considered trivial by the father or other family members, then the mother’s decision not to prioritize the baby.


*“Of course, their growth and development must be a shared concern. Especially in the 0–4-year period, children experience very rapid development both physically, cognitively, and socio-emotionally which will become a strong foundation for the family’s future.”*



*“I don’t have full authority compared to husband and in-laws.”*


### 3.8. Bathroom with Adults and Dirty

Dirty bathrooms are often associated with the nesting of viruses and bacteria. Not surprisingly, this is always associated with health problems. A dirty bathroom equals poor sanitation. For children, prolonged use can cause stunting. Stunting is closely related to chronic nutritional problems. However, if this is corrected by consuming nutritious food but not balanced with the use of a good bathroom, it will still cause stunting. Because toilets harbor various viruses and bacteria that cause various diseases, one of which is diarrhea.


*“When we bathe our children, we are always in the bathroom together with villagers who are full of garbage and dirty.”*



*“We don’t have our own bathroom, so our baby’s bathroom is shared with the adults, and it’s dirty.”*


### 3.9. FGD with Health Worker

#### Absorption by the Community

Families were always ready to take additional food packages, according to the nurse, who raised worry that since food sharing was frequent, target recipients did not always get the necessary quantity of nutritional supplements. A prominent and informed group of individuals from the community, according to the nurse, will help in ensuring that the appropriate advantages of taking dietary supplements are understood and implemented.

### 3.10. Provision of Dietary Supplements

During pandemics, nurses are worried that keeping supplement records and advising families on supplement usage would take a significant amount of time and that they will not have enough time from their normal duties to do these activities. They are also discouraged by the fact that their responsibilities are increasing, as well as the epidemic circumstances. Nurses also mentioned delays in storing supplements, which they said was the primary reason why families were unable to receive a complete supply of nutritional supplements. According to the nurses, their transportation budget is insufficient to enable them to pick up supplies on a regular basis, despite the fact that there is a growing demand for supplements from the general population.


*“It is not possible for us to get our wages and food distribution incentives on schedule.”*
(LHW)

*“A committee of prominent individuals who can ensure appropriate usage should be formed at the local level”*, says the author.


*“And please assist with distribution, else individuals would continue to demand dietary supplements for those who are not eligible.”*
(LHW)

### 3.11. When It Comes to the Supply of Dietary Supplements, There Is Occasionally a Misunderstanding

Additionally, policyholders are concerned about the documentation of the distribution and usage of nutritional supplements, and many have expressed the wish to examine these data on a frequent basis. While some health homes think that bad record keeping is linked with high workloads and pandemic circumstances, others believe that poor record-keeping is associated with inadequate nurse literacy and poor record keeping. There is limited agreement among key informants as to why financial assistance has not resulted in the regular delivery of supplements to the homes of health professionals, despite the fact that all agreed that financial help is needed. The district health office that provided nutritional supplements said that if the monitoring report was filed by the health officer, the transportation allowance would be given.


*“There should be people’s confidence in her as a nurse, but that can only be achieved if she does her responsibilities honestly, comes just for polio drops, and does not come for any other reason.”*



*“The religious leaders that mislead people by claiming that these supplements are being combined with prohibited foods/things [haraam] are a ruse by the western world to deceive people.”*



*“In addition, many recipients do not understand that the program is for their benefit; thus, there is a need to educate the general public about it.”*



*“There are some nurses who are not very good at protecting themselves. In light of the above, we must address this issue as a matter of urgency.”*



*“Despite the fact that many nurses do not submit monthly reports on time, they continue to seek timely rewards, and it is not feasible for them to get incentives if they do not submit monthly reports on time.”*


## 4. Discussion

COVID-19 has had a negative impact on many areas of business, including the health of the community. In order to understand how the COVID-19 epidemic is causing stunting, it is necessary to gather reliable data [10]. In light of the findings of the focus group, the use of dietary supplements to reduce child stunting has grown in popularity in recent years, but the available data comes from just a few well-conducted studies, making it worthwhile to conduct more research utilizing contextual process indicators. As proven in the previous research, large-scale nutritional supplementation programs may be beneficial in decreasing stunting in children if the elements that need to be most stressed by the general public and the great majority of peripheral health professionals are addressed is also covered in this study. The findings of this study reveal that the intended target group ingested the full-dose supplement for all three supplements, even when care providers lacked basic knowledge about these supplements.

The majority of people do not believe stunting to be an issue, believing it to be God’s will and a result of genetics in their families [11]. For the most part, communication between the community and the nurses is a major barrier to their ability to effectively use the intervention, and the most significant hurdle for the nurses is lack of support and time to devote to these extra tasks. The lack of public knowledge, as well as the absence of a dietary supplementing procedure, continue to be problems. Nurses do not have enough time for record-keeping and other responsibilities as a result of the pandemic circumstances increasing their workload and preventing them from being properly monitored.

Stakeholders in the district also believe that more integration of public awareness and a greater sharing of food monitoring information with public stakeholders is required. Various stakeholders, including community midwives and village elders, should be engaged in order to improve behavior change communication, guarantee the distribution of accurate information, and maintain continuous monitoring. Trials in Ghana, Haiti, Peru, Bangladesh, and Malawi revealed that dietary supplements were well accepted in these countries [5]. Evidence from Ghana and Malawi, on the other hand, showed low levels of dietary supplements in families, indicating that subsidies for the supply of dietary supplements should continue [12].

Several other studies have reported instances of supplement sharing with other household members. This highlights the cultural imperative to feed all family members as well as the maternal instinct to share food among all of her children, which may make deliberate targeting of dietary supplements more difficult to achieve in practice [13]. In addition, this highlights the cultural imperative to feed all family members, which highlights the maternal instinct to share food among all of her children. Other study has shown that variables such as the distance between collection locations, delays in funding, and delays in the distribution of supplements all have an effect on the delivery and use of supplements [14]. There is a paucity of research on the role that community health professionals play in promoting the use of dietary supplements, and there is some evidence to indicate that poor skills on the part of community workers may have a negative impact on the use of dietary supplements [15]. In the analysis of controlled trial environments in the investigation of community uptake of dietary supplements, our study’s strength lies in the fact that it is not restricted to a controlled trial environment. It is not due to a lack of expertise that the other studies on fake material on how to offer more food to youngsters generates misperceptions. Additionally, a significant prevalence of stunting is caused by the use of sweetened beverages such as sweetened condensed milk. This is due to the fact that what infants need is a diet dominant in protein. On the other hand, findings from qualitative study indicate that the reason the unclean toilet used to bathe the kid was unclean is because it is used by adults. This is the same as studies that suggests the condition that causes stunting and diarrhea are caused by a virus that is prevalent around the residence.

During the pandemic, we looked at the variables that contributed to stunting in children. We discovered that the probability of stunting in families that drink untreated water is more than three times greater than if households use improper latrines, while the likelihood of stunting in households that drink untreated processed water is 27 percent higher than if households use improper latrines. Toilet. The usage of home drinking water sources or techniques for disposing of children’s excrement were not shown to be related to stunting. Stunting was shown to be much more common in males, older children, and families with lower socioeconomic status. Stunting was shown to be considerably more common in infants aged 0–23 months whose mothers had less education; otherwise, the findings were the same as those seen in children aged 0–23 months, according to the study.

There has not been a lot of study done in Indonesia on the connection between the quality of water supply, sanitation, and nutrition [16]. Recent studies using cross-sectional survey data on the factors that contribute to stunting in children in Indonesia did not take into consideration or comment on the connection between the WASH variable and the factors that contribute to stunting [17]. Latrine density was shown to be substantially linked with better nutritional status in children aged 6–18 months in Indonesian tea plantations, according to research looking at the drivers of child development in Indonesian tea plantations [18]. This was shown to be the case in spite of the fact that the research did not discriminate between high-quality and low-quality latrines. According to the findings of an evaluation of the sanitation program in East Java, children living in households with inadequate latrines at baseline had a lower prevalence of soil-borne worms and increased height, weight, and weight-per-height; however, the effect was only significant for non-poor households, which were more likely to build latrines as a result of the program. In addition, the effect was only significant for children living in households with inadequate latrines at baseline [19]. Additionally, the effect was only significant for children living in households with inadequate latrines at baseline [20]. 

Another area of research is accumulating evidence on a connection between sanitation and stunting in low- and middle-income nations. Improved sanitation, according to an analysis of data gathered in eight nations across three continents, was shown to be substantially linked with an increase in the height of children [21]. Fink et al. [22] investigated domestic differences in stunting and sanitation using data from 172 Demographic and Health Surveys (DHS) conducted between 1986 and 2007. They found that the probability of stunting was reduced in households with access to better sanitation facilities (OR: 0.73, 95 percent CI 0.71–0.75). A recent study by Spears [23] examined cross-country variance in stunting and sanitation using data from the World Health Organization (WHO) from 65 countries and found that stunting variation explained 54 percent of the worldwide range in child height. Improved sanitation, according to a number of researchers from different nations, including cross-sectional surveys (India [23], longitudinal studies (Peru [6], and operational studies (Ethiopia), is essential for children’s linear development, among other things.

Wherever you go on the planet, people have a harder time agreeing on whether or not there is a connection between the availability of water and stunting. When information from seventy nations with low or intermediate incomes was analyzed, it was discovered that having a better water supply was connected with a decreased incidence of stunting in children. This was shown to be the case when there was a correlation between the two factors (OR: 0.92, 95 percent CI 0.89–0.94). According to the conclusions of a different piece of study, the preventive impact of improved water is dependent on the availability of other WASH elements. This is the case even when the water itself is better. Esrey made the discovery that the impact of water supply on children’s height was only beneficial among rural children when both improved sanitation and water service were provided. The data for this discovery came from eight different countries in Africa, South Asia, and South America and was collected in the 1980s. Esrey’s findings are based on these findings. This material was utilized by Esrey to support the conclusion that she came to, indicating that the protective impact of reported mother or caregiver personal hygiene behaviors was increased when they were paired with access to piped water in the household; the study was conducted in India [24]. It also showed that availability of piped water in homes increased the protective effect of the reported mother or caregiver personal hygiene habits. According to the findings of our study on the dynamic relationship between drinking water treatment and household sanitation, the treatment of domestic water has been shown to provide some protective advantages to houses that lack proper sanitation.

According to a meta-analysis of data from 14 cluster-randomized trials carried out in 10 low- and middle-income countries, there was little benefit to height in children under the age of five from WASH interventions (specifically, disinfection of water with sunlight, provision of soap, and improvement in the quality of water). These interventions included: Improving the quality of water; providing soap; and improving the quality of water [25]. The analysis is hindered by a scarcity of high-quality methodological research, particularly in the field of sanitation.

WASH facilities and behavior that are inadequate may have a negative effect on children’s nutritional condition by producing diarrhea [26], intestinal helminth infections [7], or environmental enteropathy [27]. These infections and conditions have a direct impact on nutritional status through a variety of mechanisms, including loss of appetite, loss of host tissue, maldigestion or malabsorption of nutrients, chronic immune activation, and other infection-related responses that divert nutrient and energy use, such as fever.

A study was conducted along similar lines in order to uncover the characteristics that are predictive of severe stunting in children ranging in age from 0 to 23 months. The purpose of this study was to discover the factors. Both the direction of the connection between the WASH variable and overall stunting as well as the direction of the link between the WASH variable and severe stunting followed the same pattern. However, after taking into account a number of confounding variables, the connection was not found to be significant. This might be because the frequency of severe stunting was relatively low in this cohort (6.7 percent in children aged 0–23 months and 7.7 months). In addition to a lower average family income and a lower average mother’s education level, we found that older children and males had a greater risk of stunting or severe stunting than younger children and females did. These results are in line with those discovered in studies carried out in Indonesia as well as in other low- and middle-income nations [28].

As we shall go through in the next section, there are certain shortcomings in our study. Because we used cross-sectional data, we are unable to reach to any judgments regarding whether or not there is a cause-and-effect relationship between stunting and the variables. Randomized controlled studies on the relationship between sanitation and health are difficult to carry out for a number of reasons, including the difficulty of changing people’s behaviors, the vast variety of technological demands, and the multiple channels by which excrement may be polluted [9]. The second issue is that the information on personal and family behaviors is based on the moms’ memories, which may be slanted or inaccurate. This is a concern for both the reasons [21]. In addition, there was a lack of information on the nutritional health of the mothers, the weight or length of their newborns, all of which are known to be predictors of stunting [29]. Only 823 of the infants had information on their birth weight, and the height and body mass index of the mothers were not measured. The findings of our research do, however, include indicators of the socioeconomic standing of the mother’s family as well as indicators of health-seeking behavior, both of which have the potential to add to the explanation of the mother’s nutritional and health condition. In order to complete the investigation, children who suffered from chronic diseases were excluded from the sample. This was done since it is probable that the presence of a number of chronic illnesses is linked to stunting. However, since this condition is so uncommon in babies in Indonesia who are younger than two years old, only a tiny percentage of children are being excluded from the research. Fifth, the research does not take into consideration all of the elements that impact the quality of the water used in homes, such as the correct way to handle and store water before it is used [30]. In conclusion, although the data were collected from districts that do not necessarily reflect the nation as a whole, they all display a high incidence and/or burden of stunting and reflect various types of Indonesia. This is despite the fact that these districts do not necessarily represent the nation as a whole.

This study’s findings, despite its limitations, have revealed a previously unreported relationship between stunting and factors such as household sanitation, water treatment, consumption of sweetened condensed milk, and hoax information about infant food supplements in Indonesia, according to the authors.

## 5. Conclusions

For the purpose of drawing a conclusion, our findings suggest that the state of cleanliness inside the house and the quality of the drinking water are important predictors of stunting in the population of Indonesian children aged 0 to 23 months. Concerns have also been raised over the use of sweetened condensed milk and the absence of gender parity in the decision-making procedures. The findings of this study suggest that policies and programs to combat stunting in Indonesia should devote more attention to WASH interventions in order to be effective. This is because more national and international evidence is emerging, demonstrating a link between water, sanitation, and hygiene (WASH) and linear growth in early childhood.

## Figures and Tables

**Table 1 children-09-01189-t001:** Socioeconomic characteristics of the population (*n* = 152).

	Proportion (%)
Mother’s age < 20 years	3.9
20–29 years	51.4
30–39 years	37.7
≥40 years	6.6
Mothers’ education No or incomplete primary education	14.3
Completed primary education	16.7
Completed junior high education	24.3
Completed senior high education	44.4
Number of household members ≤ 4 people	40.2
>4 people	59.6
Household water, sanitation and hygiene Improved sanitary facility	61.0
Safe disposal of child’s feces	43.3
Use of soap for handwashing	55.2
Improved source of drinking water	31.5
Treated water	90.1
Mother’s care during last pregnancy and delivery At least 4 antenatal care visits	93.2
Antenatal care provided by doctor/midwife	95.2
Antenatal care in a private or public health facility	96.4
Mother participates in household decisions Household purchases on food	88.6
What food to cook for the household	89.4
What food to give to child	95.1
Seeking health care for child	86.3
Wealth quintile Lowest	21.4
Second	20.2
Third	20.1
Fourth	18.6
Highest	19.4

**Table 2 children-09-01189-t002:** Background characteristics of children aged 0–23 months (*n* = 152 unless otherwise stated).

	Proportion (%)
Proportion (%) Sex	
Girl	48.7
Boy	51.3
Age	
0–5 months	21.7
6–11 months	24.1
12–23 months	53.5
Stunting	
Moderate stunting	20.8
Severe stunting	5.8
Total stunting	29.5
IYCF practices	
Breastfeeding initiated within one hour of birth (children 0–23 months)	68.8
Exclusive breastfeeding (children 0–5 months)	55.2
A Minimum dietary diversity of complementary food (children 6–23 months)	47.4
Minimum dietary frequency of complementary food (children 6–23 months)	72.5
Minimum acceptable diet (children 6–23 months)	35.7
Age-appropriate feeding (children 0–23 months)	41.7

**Table 3 children-09-01189-t003:** Risk factors for total stunting in children aged 0–23 months.

Factors		Stunted (%)	*n*	Unadjusted (Bivariate)			Adjusted (Multivariate)	
				OR	(95% CI)	*p*	OR	(95% CI) *p*
Sex	Boys	31.7	78	1.29	(0.99–1.68)	0.05	1.45	(1.11–1.90) 0.007
	Girls	26.9	74	1.01			1.00	
Age of child	12–23 months	38.9	96	4.02	(2.72–5.87)	<0.001	4.40	(2.97–6.53) <0.001
	6–11 months	23.7	31	1.93	(1.27–2.88)		1.97	(1.30–2.99)
	0–5 months	14.4	25	1.01			1.00wx	
Breastfeeding within one hour of birth	No	31.2	48	1.13	(0.87–1.50)	0.35		
	Yes	26.5	104	1.01				
Age-appropriate feeding	No	30.2	87	1.38	(1.09–1.77)	0.007		
	Yes	23.1	65	1.04				
Mother’s age	≥40 years	29.2	23	1.25	(0.61–2.63)	0.78		
	30–39 years	28.4	40	1.37	(0.73–2.60)			
	20–29 years	27.2	57	1.28	(0.70–2.37)			
	<20 years	25.1	33					
Mother’s education	No or incomplete primary	43.4	68	2.52	(1.77–3.62)	<0.001		
	Completed primary	31.0	19	1.47	(1.04–2.12)			
	Completed junior high	27.6	33	1.26	(0.96–1.72)			
	Completed senior high	23.0	32	1.04				
Number of household members	>4	28.9	81	1.02	(0.83–1.29)	0.75		
	≤4	27.6	71	1.01				
Wealth quintile	Lowest	40.1	53	2.83	(1.84–4.40)	<0.001	2.31	(1.43–3.68) 0.004
	Second	31.0	32	1.88	(1.33–2.70)		1.85	(1.26–2.72)
	Third	27.1	28	1.57	(1.06–2.30)		1.68	(1.11–2.55)
	Fourth	23.0	20	1.28	(0.83–1.91)		1.31	(0.85–2.04)
	Highest	19.2	19	1.06			1.04	
Sanitation	Unimproved	34.4	53	1.72	(1.37–2.15)	<0.001	1.26	(0.99–1.63) 0.07
	Improved	23.1	99	1.01			1.01	
Safe disposal of child’s feces	Unsafe	28.5	87	1.15	(0.87–1.40)	0.43		
	Safe	25.7	65	1.03				
Use of soap for hand washing	Not use soap	30.5	63	1.28	(1.00–1.67)	0.04		
	Use soap	24.7	89	1.07				
Water source	Unimproved	26.2	93	0.87	(0.67–1.10)	0.24		
	Improved	29.9	59	1.02				
Water treatment	Untreated	37.2	27	1.57	(1.08–2.34)	0.018	0.87	(0.49–1.61) 0.70
	Treated	26.4	125	1.01			1.02	
Number of ANC visits of mother during	<4	41.8	15	1.71	(1.12–2.60)	0.012		
pregnancy	≥4	26.6	137	1.01				
Doctor/midwife provided ANC to mother	No	46.4	34	2.08	(1.29–3.33)	0.001		
during pregnancy	Yes	28.5	118	1.04				
ANC in private or public health facility	No	46.9	48	2.13	(1.16–3.87)	0.012		
	Yes	26.8	104	1.01				
Mother participates in decisions on	Yes	27.7	121	1.22	(0.78–1.87)	0.38		
Mother participates in decisions on what food is cooked for HH	Yes	28.2	85	1.51	(0.95–2.38)	0.07		
	No	20.7	67	1.02				
Mother participates in decisions on food	Yes	27.5	130	1.19	(0.61–2.27)	0.62		
given to child	No	26.8	22	1.05				
Mother participates in decisions on seeking	Yes	27.7	118	1.16	(0.79–1.61)	0.51		
health care for child	No	25.3	34	1.03				
Sanitation x water treatment								2.71 (1.32–5.97) 0.007

**Table 4 children-09-01189-t004:** Theme results focus group discussion.

Respondent FGD	Theme
women who have children that are stunting	Underestimating the importance of dietary supplements
	Having a fear of using dietary supplements
	Frequent Shopping Doesn’t Matter
	Children eat sweet food
	Following the wrong Guide
	Women’s involvement in policymaking, decision making, and control
	Bathroom with adults and dirty
Health Worker	Absorption by the community
	Provision of dietary supplements
	When it comes to the supply of dietary supplements, there is occasionally a misunderstanding.

## Data Availability

On request, data will be presented to each individual reader. Reader may request by email.

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
