# Peer review of "Determining the Factors That Influence Stunting during Pandemic in Rural Indonesia: A Mixed Method"

_children, 2022, doi:10.3390/children9081189_

Round 1
Reviewer 1 Report
Abstract
- the methods section does not indicate what was done for the quantitative and qualitative components and the type of analysis that was conducted.
- the results do not indicate the level of significance?
- “In addition, the usage of sweetened condensed milk from government grants, the absence of gender equity in the decision-making process, and the link between WASH and linear development in early infancy must be explored.” This statement seems like a reiteration of what was done by this study rather than a recommendation.
Introduction
- Page 2, line 3- “Authorities should consider the direct health consequences of the pandemic and the indirect consequences and responses to the epidemic while assessing their choices”- this statement is clear
- Page 2, line5- “though the death rate for COVID-19 seems to be modest among children and women of reproductive age, the virus has been linked to several other diseases [3]” – such as?
- During the COVID-19 pandemic, we wanted to see what kind of indirect effects there could be from stunting- I suggest you move this to a later part of the introduction.
- Page 2, paragraph 2- needs revision, link the statements
- Page 2, paragraph 3
- include the reference for the statement: “Since the pandemic period in 2020, the poverty rate has increased”, “Experts also said that the stunting rate from 27 percent would increase to 32 percent.”, and “ This shows that the COVID-19 pandemic does have an effect on stunting”.
- I was not able to understand this statement-“If he affects the economy, of course the follow-up impacts will be many, including the birth of stunting babies” – I suggest you edit it.
The introduction needs major revision. The above are some examples.
Materials and Method
- Page 2, the first line of the methods section- “We performed the research between 9-31 Maret 2021” – what is Maret?
- Page 3, the first line- “a dry seasonal environment (906 mm average annual rainfall) (906 mm average annual rainfall”- repeated
- Page 3, line 2- “The mixed-methods technique comprises three components.”- what were the components?
- Page 3, line 2 -“Objectives: (1) discussion groups with caregiver and mothers, (2) distributing questionnaires to all homes without sample, and (3) anthropometric measurements of children without sampling.”- what do you mean? Is this the objective of the study?
- Page 3, line 4- “We work with nurses and the government as an intercultural team to design research tools that reflect the local environment and promote participant participation in the project. Nurses and the government selected the study subject (e.g., stunting) with feedback from several communities. We included these questions in the questionnaire.”- I suggest you provide details about the questionnaire development, pretest, and adaptation to local context if any?
- Subject
o In the subject selection, you first mention “sample that will be used in this study is 152 mothers who have children aged 24-59 months” then later in the inclusion you have a statement “Mothers who have a child aged 0-35 months, 2) Children aged 0-35 months who have MCH/KMS and are registered at the Puskesmas Pasirjati, Bandung.
- Sampling
o How were clusters selected? And I am not sure what the relevance of the random number generator was? Did you have a sampling frame? If so it would be good to clearly explain that.
- Data collection
o What do you mean by “systematic questionnaire used”?
- Data input and analysis are required
o What does the heading imply?
o It is good to indicate the steps taken for cleaning, and analyzed separately. I also recommend that you have a section explaining the operational definitions.
o What was done for the qualitative data?
-
Author Response
Response to the Reviewer 1
|
No. |
Comment |
Response |
Modification |
Line No./ Page |
|
1 |
Abstract 1. the methods section does not indicate what was done for the quantitative and qualitative components and the type of analysis that was conducted. 2. the results do not indicate the level of significance? 3. - “In addition, the usage of sweetened condensed milk from government grants, the absence of gender equity in the decision-making process, and the link between WASH and linear development in early infancy must be explored.” This statement seems like a reiteration of what was done by this study rather than a recommendation. |
1. Quantitative using a baseline survey which was analyzed using Stata 11.0. focus group discussions which were analyzed using Nvivo 12 with a questionnaire, and anthropometric measurements of children from surveyed households. 2. significant because in the bivariate test the value of male gender has a number of 0.05 |
1. We use mixed methods. The respondents of this study were 152 mothers for the Maternal and Child Nutrition Security project, the sampling technique is Cluster Sampling. Quantitative using a baseline survey which was analyzed using Stata 11.0. The qualitative data used focus group discussions which were analyzed using Nvivo 12 with a questionnaire, and anthropometric measurements of children from surveyed households. 2. The need for further research related to government assistance related to improving toddler nutrition, as well as the relationship between WASH and linear development in early infancy should be explored |
Line 53-57, Page 2
Line 67-69, Page 2
|
|
2 |
Introduction 1. Page 2, line 3- “Authorities should consider the direct health consequences of the pandemic and the indirect consequences and responses to the epidemic while assessing their choices”- this statement is clear 2. Page 2, line5- “though the death rate for COVID-19 seems to be modest among children and women of reproductive age, the virus has been linked to several other diseases [3]” – such as? 3. During the COVID-19 pandemic, we wanted to see what kind of indirect effects there could be from stunting- I suggest you move this to a later part of the introduction. 4. Page 2, paragraph 2- needs revision, link the statements 5. Page 2, paragraph 3 6. include the reference for the statement: “Since the pandemic period in 2020, the poverty rate has increased”, “Experts also said that the stunting rate from 27 percent would increase to 32 percent.”, and “ This shows that the COVID-19 pandemic does have an effect on stunting”. 7. I was not able to understand this statement-“If he affects the economy, of course the follow-up impacts will be many, including the birth of stunting babies” – I suggest you edit it. |
4. we move this to the next introductory section.
6. we added references |
2. Although the mortality rate for COVID-19 appears modest among children and women of reproductive age, the virus has been linked to several diseases such as dengue fever, tuberculosis, measles.
5. During the COVID-19 pandemic, we wanted to see what the indirect effects of stunting might look like. In addition, multi-sectoral operations have not been carried out optimally in Indonesia due to the lack of information about the causes of stunting in Indonesia during the pandemic which can be used as guidelines for designing multi-sectoral programs. Specifically on stunting, the World Health Organization (WHO) has published two policy briefs that guide how to accelerate progress towards global targets and combat stunting while ensuring equity [2]. as action points to realize the stunting reduction agenda. 7. Since the pandemic period in 2020, the poverty rate has increased, of course there will be many further impacts, including the birth of stunting babies. Experts also say that the stunting rate from 27 percent will increase to 32 percent |
Line 78-80, Page 3
Line 81-87, Page 3
Line 88-90, Page 3
|
|
3 |
Materials and Method - Page 2, the first line of the methods section- “We performed the research between 9-31 Maret 2021” – what is Maret? - Page 3, the first line- “a dry seasonal environment (906 mm average annual rainfall) (906 mm average annual rainfall”- repeated - Page 3, line 2- “The mixed-methods technique comprises three components.”- what were the components? - Page 3, line 2 -“Objectives: (1) discussion groups with caregiver and mothers, (2) distributing questionnaires to all homes without sample, and (3) anthropometric measurements of children without sampling.”- what do you mean? Is this the objective of the study? - Page 3, line 4- “We work with nurses and the government as an intercultural team to design research tools that reflect the local environment and promote participant participation in the project. Nurses and the government selected the study subject (e.g., stunting) with feedback from several communities. We included these questions in the questionnaire.”- I suggest you provide details about the questionnaire development, pretest, and adaptation to local context if any? - Subject o In the subject selection, you first mention “sample that will be used in this study is 152 mothers who have children aged 24-59 months” then later in the inclusion you have a statement “Mothers who have a child aged 0-35 months, 2) Children aged 0-35 months who have MCH/KMS and are registered at the Puskesmas Pasirjati, Bandung. - Sampling o How were clusters selected? And I am not sure what the relevance of the random number generator was? Did you have a sampling frame? If so it would be good to clearly explain that. - Data collection o What do you mean by “systematic questionnaire used”? - Data input and analysis are required o What does the heading imply? o It is good to indicate the steps taken for cleaning, and analyzed separately. I also recommend that you have a section explaining the operational definitions. o What was done for the qualitative data? |
We delete the repeated sentences
we mean not the objective, but the mixed-methods technique
We have written a systematic explanation of the questionnaire in the data collection section
we mean age 0-23 months, sorry for this mistake
systematic questionnaire is a detailed explanation related to the questionnaire,
we explain about qualitative data in the section on qualitative analytical statistics |
March
The mixed-methods technique comprises three components : (1) discussion groups with caregiver and mothers, (2) distributing questionnaires to all homes without sample, and (3) anthropometric measurements of children without sampling
Pasirjati has 14 clusters. In the first stage, sub-districts (called clusters) were selected in each cluster, in the second stage the Puskesmas (Community Health Services) were randomly selected from each sub-district and finally, villages were randomly selected from the Puskesmas. The selection of households in each cluster was taken randomly using a sampling frame of every 10 households with the nearest household from the village health service (Pustu) as the starting point. The number of samples that have been calculated consists of, cluster 1 (9 samples), cluster 2 (12 samples), cluster 3 (12 samples), cluster 4 (10 samples), cluster 5 (10 samples), cluster 6 (12 samples), cluster 7 (14 samples), cluster 8 (10 samples), cluster 9 (11 samples), cluster 10 (9 samples), cluster 11 (11 samples), cluster 12 (10 samples), cluster 13 (13 samples) and cluster 14 (11 samples). The use of Cluster Sampling is intended so that each cluster has a representative. Sample selection was assisted by using a Random Number Generator (RNG) where the data had previously been sorted according to the research criteria and grouped based on the cluster where the respondent lived. This RNG is used to facilitate researchers in selecting respondents randomly.
Performing a Stunt (High-for-age) Anthropometric measurements were used to assess the nutritional condition of children under the age of five. Children less than two years old had their length measured, while those more than two years old had their height measured. A wooden stadiometer and a Microtoice tape were used to measure length and height, respectively, to within 0.1 cm. The standard deviation (SD) (Z-score) from the median of the reference population is used to indicate the current state of the measurement of height according to age. A kid was termed stunted if their standard deviation in height (SD) was more than or equal to two standard deviations below the median of the reference population. Socio-economic aspects A structured household questionnaire was used to collect data on the following family-level factors: region (urban and rural), district (eight in total), father's education level (completed primary school [6 years of schooling], completed secondary school years of schooling, and completed secondary school [12 years of schooling]); mother's education level; parental education (both with higher education, father with higher education, mother with higher education); and child's education level (completed primary school [6 years of schooling], completed In addition, the following child-level characteristics were aggregated: the child's age in months, gender, the provision of information on nutritional status throughout pregnancy, and the number of prenatal visits. The questionnaire was conducted after the collection of written informed permission. The field supervisor examined the surveys daily for correctness, consistency, and completeness.
Analytical statistics Using the EPIINFO data entry application, data were input into a computerized database and sanitized [9]. Data on nutritional status were examined using a new World Health Organization growth reference. The household ownership of the aforementioned consumer items was used to calculate the wealth index score using a technique similar to that outlined by Filmer and Pritchett and then categorized into three groups. The lowest forty percent of families are referred to as the poorest, the next forty percent as middle class, and the top twenty percent as the poorest. Analysis of moderate and severe stunting in infants and toddlers aged 0 to 23 months. The dependent variable was stated as a dichotomous variable to identify the amount of stunting and severe stunting: category 0 if not stunted (-2SD) or severely stunted (-3SD) and category 1 if stunted (-2SD) or extremely stunted (-3SD). First, a univariate binary logistic regression analysis was undertaken to assess the link between stunted and severely stunted 0-23-month-old children. In a second step, a multivariate logistic regression model was used to analyze the variables associated with stunting and severe stunting. A step-by-step strategy to reverse elimination is used. The initial variable was selected for inclusion in the model if the univariate p value was less than or equal to 0.25. In the final model, only the factors that were significantly linked with stunted and extremely stunted children (p 0.05) remained. The logistic model's unadjusted and adjusted odds ratios are shown with confidence intervals of 95 percent. For data analysis, the 'SVY' command from Stata version 11 (Stata Corp.) was used to alter the cluster sampling design and sample weights appropriately. |
Line 119, Page 4
Line 121-123, Page 4
Line 142-154, Page 4-5
Line 171-269, Page 5-6
|

Reviewer 2 Report
The introduction reads well and provides basic information about the possibility of pandemics and child stunting.
The method details the step-by-step conduct of the study, which may allow for replication of the study if needed. It is great to see that triangulation was used to strengthen the findings. Although a few things need to be made clearer. Detail below.
The results were adequately discussed and the conclusion supports the key findings of the study.
Comments
Introduction
Please note that COVID-19 is the pandemic and SARS-Cov/coronavirus is the causative agent. Ensure correct use of terminology e.g. in the first paragraph of your introduction; severe acute respiratory syndrome coronavirus IS NOT an abbreviation for COVID-19.
Please check the last sentence of your 2nd paragraph in the introduction. There may be inappropriate punctuation/incomplete sentence
…while ensuring equity [2]. as action points for putting a stunting reduction agenda into action.
Method
Line 3 under ‘subject’ sub-heading;
Fourteen districts were selected ‘to represents’ instead of ‘for represents’…
Please re-check the index child age considered in the survey and make clearer. Two age groups appeared; …research sample that will be used in this study is 152 mothers who have children aged 24-59 months. Then under inclusion Criteria; Mothers who have a child aged 0-35 months.
Could you briefly explain what informed the selection of the sample size 152? E.g. A previous study, validated sample size formula, national representative data or power analysis?. Also, how many of the participants were considered for the focus group and how many participated? Are the FGD participants a subset of the quantitative survey? These need to be clearly stated.
Result
Check the result description in table 3 for tautology;
…Stunting was more prevalent in children whose mothers did not complete basic education (43.4 percent) or did not complete basic education but did not complete basic education but did not complete basic education (31.0 percent )…
Also, recheck the confidence interval in the phrase below. I wonder if the upper CI is higher than stated. It is usually unlikely for the OR to be exactly the same as the upper CI, particularly when the result is significant;
…The AOR 1.28, 95 percent confidence interval 1.04-1.28 for stunting in households that treated their own water was found…
I was initially wondering how the interaction between ‘Household water, sanitation and hygiene Improved sanitary facility’ was determined. I was expecting the individual variable data in table 1 and the interaction to be computed in the regression model. Thereafter, I realised it was referring to WASH as a variable. Please make it clear in the method that the WASH variable was collected as such, otherwise you need to state and demonstrate how you determine the interaction effect.
Author Response
Response to the Reviewer 2
|
No. |
Comment |
Response |
Modification |
Line No./ Page |
|
1 |
Introduction Please note that COVID-19 is the pandemic and SARS-Cov/coronavirus is the causative agent. Ensure correct use of terminology e.g. in the first paragraph of your introduction; severe acute respiratory syndrome coronavirus IS NOT an abbreviation for COVID-19. Please check the last sentence of your 2nd paragraph in the introduction. There may be inappropriate punctuation/incomplete sentence …while ensuring equity [2]. as action points for putting a stunting reduction agenda into action. |
we added in the article.
Sorry, not punctuation but a comma in the sentence |
1. Communities around the world are taking steps to limit the spread of the acute respiratory syndrome coronavirus or CoronaVirus Disease-2019 (COVID-19) and reduce the number of deaths caused by the virus [1]. |
Line 79-81, Page 2
|
|
2 |
Method Line 3 under ‘subject’ sub-heading; Fourteen districts were selected ‘to represents’ instead of ‘for represents’… Please re-check the index child age considered in the survey and make clearer. Two age groups appeared; …research sample that will be used in this study is 152 mothers who have children aged 24-59 months. Then under inclusion Criteria; Mothers who have a child aged 0-35 months.
Could you briefly explain what informed the selection of the sample size 152? E.g. A previous study, validated sample size formula, national representative data or power analysis?. Also, how many of the participants were considered for the focus group and how many participated? Are the FGD participants a subset of the quantitative survey? These need to be clearly stated. |
We have replaced according to your suggestion, for the age group, only 0-23 months old
underlying the selection of the sample size 152 due to the validated sample size formula, and power analysis
We have explained the focus of the discussion group in the Quantitative and Qualitative analytical statistics section |
|
|
|
3 |
Result Check the result description in table 3 for tautology; …Stunting was more prevalent in children whose mothers did not complete basic education (43.4 percent) or did not complete basic education but did not complete basic education but did not complete basic education (31.0 percent )… Also, recheck the confidence interval in the phrase below. I wonder if the upper CI is higher than stated. It is usually unlikely for the OR to be exactly the same as the upper CI, particularly when the result is significant; …The AOR 1.28, 95 percent confidence interval 1.04-1.28 for stunting in households that treated their own water was found…
I was initially wondering how the interaction between ‘Household water, sanitation and hygiene Improved sanitary facility’ was determined. I was expecting the individual variable data in table 1 and the interaction to be computed in the regression model. Thereafter, I realised it was referring to WASH as a variable. Please make it clear in the method that the WASH variable was collected as such, otherwise you need to state and demonstrate how you determine the interaction effect.
|
we added in the article
Sorry for the error, indeed it is not 1.28 but 1.29. we have replaced in the table and results
Our introduction explains that WASH has indicators of water, sanitation and household hygiene facilities. |
Stunting was more common in children whose mothers did not complete basic education (43.4 percent) than mothers who completed basic education (31.0 percent) |
|
